# Predicting Cellular Responses with Variational Causal Inference and Refined Relational Information

**Yulun Wu**
University of California, Berkeley
yulun_wu@berkeley.edu

**Robert A. Barton**
Immunai
robert.barton@immunai.com

**Zichen Wang**
Amazon
zichewan@amazon.com

**Vassilis N. Ioannidis**
Amazon
ivasilei@amazon.com

**Carlo De Donno**
Immunai
carlo.dedonno@immunai.com

**Layne C. Price**
Amazon
prilayne@amazon.com

**Luis F. Voloch**
Immunai
luis@immunai.com

**George Karypis**
Amazon
gkarypis@amazon.com

## Abstract

Predicting the responses of a cell under perturbations may bring important benefits to drug discovery and personalized therapeutics. In this work, we propose a novel graph variational Bayesian causal inference framework to predict a cell's gene expressions under counterfactual perturbations (perturbations that this cell did not factually receive), leveraging information representing biological knowledge in the form of gene regulatory networks (GRNs) to aid individualized cellular response predictions. Aiming at a data-adaptive GRN, we also developed an adjacency matrix updating technique for graph convolutional networks and used it to refine GRNs during pre-training, which generated more insights on gene relations and enhanced model performance. Additionally, we propose a robust estimator within our framework for the asymptotically efficient estimation of marginal perturbation effect, which is yet to be carried out in previous works. With extensive experiments, we exhibited the advantage of our approach over state-of-the-art deep learning models for individual response prediction.

## 1 Introduction

Studying a cell's response to genetic, chemical, and physical perturbations is fundamental in understanding various biological processes and can lead to important applications such as drug discovery and personalized therapies. Cells respond to exogenous perturbations at different levels, including epigenetic (DNA methylation and histone modifications), transcriptional (RNA expression), translational (protein expression), and post-translational (chemical modifications on proteins). The availability of single-cell RNA sequencing (scRNA-seq) datasets has led to the development of several methods for predicting single-cell transcriptional responses (Ji et al., 2021). These methods fall into two broad categories. The first category (Lotfollahi et al., 2019; 2020; Rampášek et al., 2019; Russkikh et al., 2020; Lotfollahi et al., 2021a) approaches the problem of predicting single cell gene expression response without explicitly modeling the gene regulatory network (GRN), which is widely hypothesized to be the structural causal model governing transcriptional responses of cells (Emmert-Streib et al., 2014). Notably among those studies, CPA (Lotfollahi et al., 2021a) uses an adversarial autoencoder framework designed to decompose the cellular gene expression response to latent components representing perturbations, covariates and basal cellular states. CPA extends the classic idea of decomposing high-dimensional gene expression response into perturbation vectors (Clark et al., 2014; 2015), which can be used for finding connections among perturbations (Subramanian et al., 2017). However, while CPA's adversarial approach encourages latent indepen-

dence, it does not have any supervision on the counterfactual outcome construction and thus does not explicitly imply that the counterfactual outcomes would resemble the observed response distribution. Existing self-supervised counterfactual construction frameworks such as GANITE (Yoon et al., 2018) also suffer from this problem.

The second class of methods explicitly models the regulatory structure to leverage the wealth of the regulatory relationships among genes contained in the GRNs (Kamimoto et al., 2020). By bringing the benefits of deep learning to graph data, graph neural networks (GNNs) offer a versatile and powerful framework to learn from complex graph data (Bronstein et al., 2017). GNNs are the *de facto* way of including relational information in many health-science applications including molecule/protein property prediction (Guo et al., 2022; Ioannidis et al., 2019; Strokach et al., 2020; Wu et al., 2022a; Wang et al., 2022), perturbation prediction (Roohani et al., 2022) and RNA-sequence analysis (Wang et al., 2021). In previous work, Cao & Gao (2022) developed GLUE, a framework leveraging a fine-grained GRN with nodes corresponding to features in multi-omics datasets to improve multimodal data integration and response prediction. GEARS (Roohani et al., 2022) uses GNNs to model the relationships among observed and perturbed genes to predict cellular response. These studies demonstrated that relation graphs are informative for predicting cellular responses. However, GLUE does not handle perturbation response prediction, and GEARS's approach to randomly map subjects from the control group to subjects in the treatment group is not designed for response prediction at an individual level (it cannot account for heterogeneity of cell states).

GRNs can be derived from high-throughput experimental methods mapping chromosome occupancy of transcription factors, such as chromatin immunoprecepitation sequencing (ChIP-seq), and assay for transposase-accessible chromatin using sequencing (ATAC-seq). However, GRNs from these approaches are prone to false positives due to experimental inaccuracies and the fact that transcription factor occupancy does not necessarily translate to regulatory relationships (Spitz & Furlong, 2012). Alternatively, GRNs can be inferred from gene expression data such as RNA-seq (Maetschke et al., 2014). It is well-accepted that integrating both ChIP-seq and RNA-seq data can produce more accurate GRNs (Mokry et al., 2012; Jiang & Mortazavi, 2018; Angelini & Costa, 2014). GRNs are also highly context-specific: different cell types can have very distinctive GRNs mostly due to their different epigenetic landscapes (Emerson, 2002; Davidson, 2010). Hence, a GRN derived from the most relevant biological system is necessary to accurately infer the expression of individual genes within such system.

In this work, we employed a novel variational Bayesian causal inference framework to construct the gene expressions of a cell under counterfactual perturbations by explicitly balancing individual features embedded in its factual outcome and marginal response distributions of its cell population. We integrated a gene relation graph into this framework, derived the corresponding variational lower bound and designed an innovative model architecture to rigorously incorporate relational information from GRNs in model optimization. Additionally, we propose an adjacency matrix updating technique for graph convolutional networks (GCNs) in order to impute and refine the initial relation graph generated by ATAC-seq prior to training the framework. With this technique, we obtained updated GRNs that discovered more relevant gene relations (and discarded insignificant gene relations in this context) and enhanced model performance. Besides, we propose an asymptotically efficient estimator for estimating the average effect of perturbations under a given cell type within our framework. Such marginal inference is of great biological interest because scRNA-seq experimental results are typically averaged over many cells, yet robust estimations have not been carried out in previous works on predicting cellular responses.

We tested our framework on three benchmark datasets from Srivatsan et al. (2020), Schmidt et al. (2022) and a novel CROP-seq genetic knockout screen that we release with this paper. Our model achieved state-of-the-art results on out-of-distribution predictions on differentially-expressed genes — a task commonly used in previous works on perturbation predictions. In addition, we carried out ablation studies to demonstrate the advantage of using refined relational information for a better understanding of the contributions of framework components.

## 2 PROPOSED METHOD

In this section we describe our proposed model — Graph Variational Causal Inference (graphVCI), and a relation graph refinement technique. A list of all notations can be found in Appendix A.

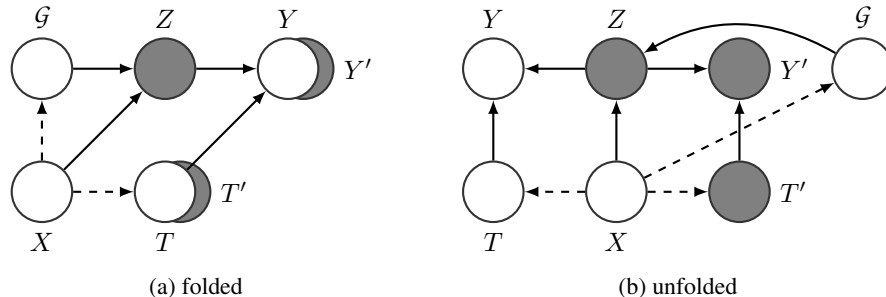

(a) folded                    (b) unfolded

Figure 1: The causal relation diagram. Each individual has a feature state $Z$ following a conditional distribution $p(Z|\mathcal{G}, X)$. Treatment $T$ (or counterfactual treatment $T'$) along with $Z$ determines outcome $Y$ (or counterfactual outcome $Y'$). In the causal diagram, white nodes are observed and dark grey nodes are unobserved; dashed relations are optional (case dependant). In the context of this paper, graph $\mathcal{G}$ is a deterministic variable that is invariant across all individuals.

## 2.1 COUNTERFACTUAL CONSTRUCTION FRAMEWORK

We define outcome $Y : \Omega \to \mathbb{R}^n$ to be a $n$-dimensional random vector (e.g. gene expressions), $X : \Omega \to E_X$ to be a $m$-dimensional mix of categorical and real-valued covariates (e.g. cell types, donors, etc.), $T : \Omega \to E_T$ to be a $r$-dimensional categorical or real-valued treatment (e.g. drug perturbation) on a probability space $(\Omega, \Sigma, P)$. We seek to construct an individual's counterfactual outcome under counterfactual treatment $a \in E_T$ from two major sources of information. One source is the individual features embedded in high-dimensional outcome $Y$. The other source is the response distribution of similar subjects (subjects that have the same covariates as this individual) that indeed received treatment $a$.

We employ the variational causal inference (Wu et al., 2022b) framework to combine these two sources of information. In this framework, a covariate-dependent feature vector $Z : \Omega \to \mathbb{R}^d$ dictates the outcome distribution along with treatment $T$; counterfactuals $Y'$ and $T'$ are formulated as separate variables apart from $Y$ and $T$ with a conditional outcome distribution $p(Y'|Z, T' = a)$ identical to its factual counterpart $p(Y|Z, T = a)$ on any treatment level $a$. The learning objective is described as a combined likelihood of individual-specific treatment effect $p(Y'|Y, X, T, T')$ (first source) and the traditional covariate-specific treatment effect $p(Y|X, T)$ (second source):

$$J(D) = \log\left[p(Y'|Y, X, T, T')\right] + \log\left[p(Y|X, T)\right] \tag{1}$$

where $D = (X, Z, T, T', Y, Y')$. Additionally, we assume that there is a graph structure $\mathcal{G} = (\mathcal{V}, \mathcal{E})$ that governs the relations between the dimensions of $Y$ through latent $Z$, where $\mathcal{V} \in \mathbb{R}^{n \times v}$ is the node feature matrix and $\mathcal{E} \in \{0, 1\}^{n \times n}$ is the node adjacency matrix. For example, in the case of single-cell perturbation dataset where $Y$ is the expression counts of genes, $\mathcal{V}$ is the gene feature matrix and $\mathcal{E}$ is the GRN that governs gene relations. See Figure 1 for a visualization of the causal diagram. The objective under this setting is thus formulated as

$$J(\mathcal{D}) = \log\left[p(Y'|Y, \mathcal{G}, X, T, T')\right] + \log\left[p(Y|\mathcal{G}, X, T)\right] \tag{2}$$

where $\mathcal{D} = (\mathcal{G}, D)$. The counterfactual outcome $Y'$ is always unobserved, but the following theorem provides us a roadmap for the stochastic optimization of this objective.

**Theorem 1.** *Suppose that* $\mathcal{D} = (\mathcal{G}, X, Z, T, T', Y, Y')$ *follows a causal structure defined by the Bayesian network in Figure 1. Then* $J(\mathcal{D})$ *has the following evidence lower bound:*

$$J(\mathcal{D}) \geq \mathbb{E}_{p(Z|Y,\mathcal{G},X,T)} \log\left[p(Y|Z, T)\right] + \log\left[p(Y'|\mathcal{G}, X, T')\right]$$
$$- \text{KL}\left[p(Z|Y, \mathcal{G}, X, T) \| p(Z|Y', \mathcal{G}, X, T')\right]. \tag{3}$$

Proof of the theorem can be found in Appendix B. We estimate $p(Z|Y, \mathcal{G}, X, T)$ and $p(Y|Z, T)$ (as well as $p(Y'|Z, T')$) with a neural network encoder $q_\phi$ and decoder $p_\theta$, and optimize the following weighted approximation of the variational lower bound:

$$J(\theta, \phi) = \mathbb{E}_{q_\phi(Z|Y,\mathcal{G},X,T)} \log\left[p_\theta(Y|Z, T)\right] + \omega_1 \cdot \log\left[\hat{p}(\tilde{Y}'_{\theta,\phi}|\mathcal{G}, X, T')\right]$$
$$- \omega_2 \cdot \text{KL}\left[q_\phi(Z|Y, \mathcal{G}, X, T) \| q_\phi(Z|\tilde{Y}'_{\theta,\phi}, \mathcal{G}, X, T')\right] \tag{4}$$

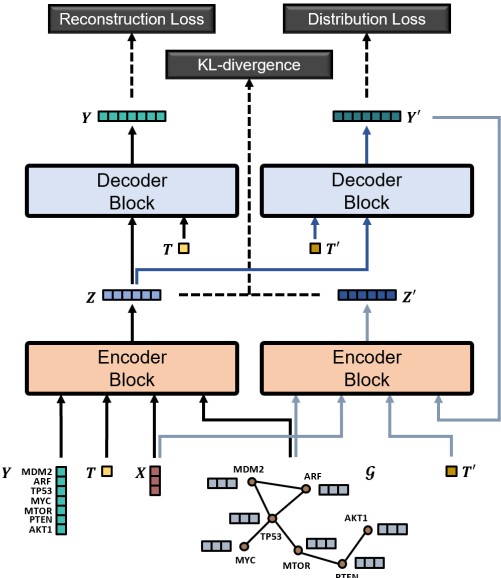

Figure 2: Model workflow — variational causal perspective. In a forward pass, the graphVCI encoder takes graph $\mathcal{G}$ (e.g. gene relation graph), outcome $Y$ (e.g. gene expressions), covariates $X$ (e.g. cell types, donors, etc.) and treatment $T$ (e.g. drug perturbation) as inputs and generates latent $Z$; $(Z, T)$ and $(Z, T')$ where $T'$ is a randomly sampled counterfactual treatment are separately passed into the graphVCI decoder to attain reconstruction of $Y$ and construction of counterfactual outcome $Y'$; $Y'$ is then passed back into the encoder along with $\mathcal{G}$, $X$, $T'$ to attain counterfactual latent $Z'$. The objective consists of the reconstruction loss of $Y$, the distribution loss of $Y'$ and the KL-divergence between the conditional distributions of $Z$ and $Z'$.

where $\omega_1$, $\omega_2$ are scaling coefficients; $\tilde{Y}'_{\theta,\phi} \sim E_{q_\phi(Z|Y,\mathcal{G},X,T)} p_\theta(Y'|Z,T')$ and $\hat{p}$ is the covariate-specific model fit of the outcome distribution (notice that $p(Y'|\mathcal{G}, X, T' = a) = p(Y|\mathcal{G}, X, T = a)$ for any $a$). In our case where covariates are limited and discrete (cell types and donors), we simply let $\hat{p}(Y|\mathcal{G}, X, T)$ be the smoothened empirical distribution of $Y$ stratified by $X$ and $T$ (notice that $\mathcal{G}$ is fixed across subjects). Generally, one can train a discriminator $\hat{p}(\cdot|Y, \mathcal{G}, X, T)$ with the adversarial approach (Goodfellow et al., 2014) and use $\log\left[\hat{p}(1|\tilde{Y}'_{\theta,\phi}, \mathcal{G}, X, T')\right]$ for $\log\left[\hat{p}(\tilde{Y}'_{\theta,\phi}|\mathcal{G}, X, T')\right]$ if $p(Y|\mathcal{G}, X, T)$ is hard to fit. See Figure 2 for an overview of the model structure. Note that the decoder estimates the conditional outcome distribution of $Y'$, in which case $T'$ need not necessarily be sampled according to a certain true distribution $p(T'|X)$ during optimization.

We refer to the negative of the first term in Equation 4 as reconstruction loss, the negative of the second term as distribution loss, and the positive of the third term as KL-divergence. As discussed in Wu et al. (2022b), the negative KL-divergence term in the objective function encourages the preservation of individuality in counterfactual outcome constructions.

**Marginal Effect Estimation** Although perturbation responses at single cell resolution offers microscopic view of the biological landscape, oftentimes it is fundamental to estimate the average population effect of a perturbation in a given cell type. Hence in this work, we developed a robust estimation for the causal parameter $\Psi(p) = \mathbb{E}_p(Y'|X = c, T' = a)$ — the marginal effect of treatment $a$ within a covariate group $c$. We propose the following estimator that is asymptotically efficient when $\mathbb{E}_{p_\theta}(Y'|\tilde{Z}_{k,\phi}, T'_k = a)$ is estimated consistently and some other regularity conditions (Van Der Laan & Rubin, 2006) hold:

$$\hat{\Psi}_{\theta,\phi} = \frac{1}{n_{a,c}} \sum_{k=1_{a,c}}^{n_{a,c}} \left\{ Y_k - \mathbb{E}_{p_\theta}(Y'|\tilde{Z}_{k,\phi}, T'_k = a) \right\} + \frac{1}{n_c} \sum_{k=1_c}^{n_c} \left\{ \mathbb{E}_{p_\theta}(Y'|\tilde{Z}_{k,\phi}, T'_k = a) \right\} \quad (5)$$

where $(Y_k, X_k, T_k)$ are the observed variables of the $k$-th individual; $\tilde{Z}_{k,\phi} \sim q_\phi(Z|Y_k, \mathcal{G}, X_k, T_k)$; $(1_c, \ldots, n_c)$ are the indices of the observations having $X = c$ and $(1_{a,c}, \ldots, n_{a,c})$ are the indices

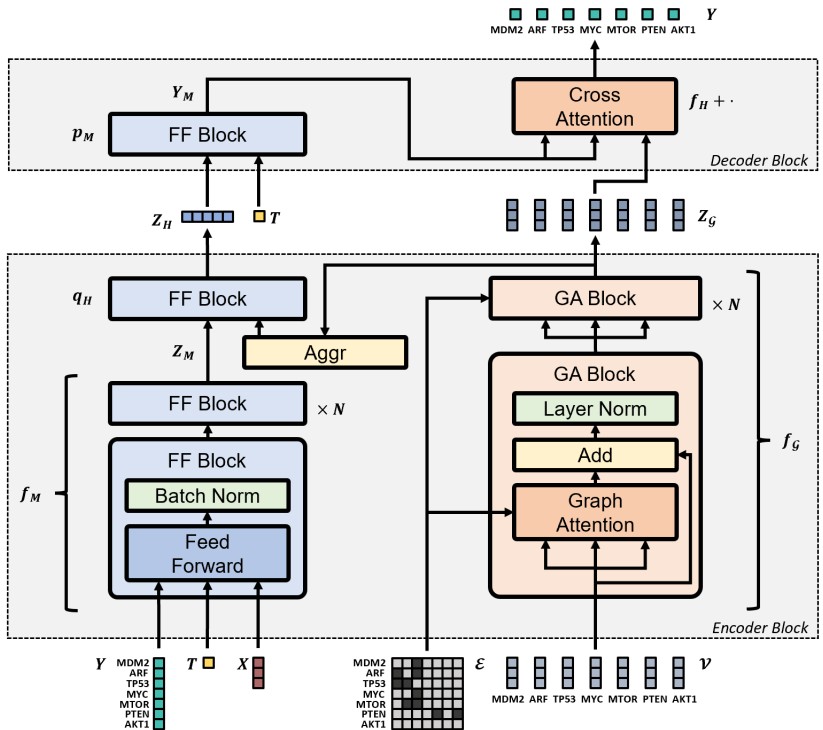

Figure 3: Model architecture — graph attentional perspective. Structure of the graphVCI encoder and decoder defined by Equations (6) to (11). Note that in the case of single-cell perturbation datasets, the graph inputs are fixed across samples and graph attention can essentially be reduced to weighted graph convolution.

of the observations having both $T = a$ and $X = c$. The derivation of this estimator can be found in Appendix C.2 and some experiment results can be found in Appendix C.1. Intuitively, we use the prediction error of the model on the observations that indeed have covariate level $c$ and received treatment $a$ to adjust the empirical mean. Prior works in cellular response studies only use the empirical mean estimator over model predictions to estimate marginal effects.

## 2.2 INCORPORATING RELATIONAL INFORMATION

Since the elements in the outcome $Y$ are not independent, we aim to design a framework that can exploit predefined relational knowledge among them. In this section, we demonstrate our model structure for encoding and aggregating relation graph $\mathcal{G}$ within the graphVCI framework. Denote deterministic models as $f_.$, probabilistic models (output probability distributions) as $q_.$ and $p_.$. We construct feature vector $Z$ as an aggregation of two latent representations:

$$Z = (Z_\mathcal{G}, Z_H) \tag{6}$$

$$Z_H \sim q_H\left(Z_M, \text{aggr}_\mathcal{G}(Z_\mathcal{G})\right) \tag{7}$$

$$Z_M = f_M(Y, X, T) \tag{8}$$

$$Z_\mathcal{G} = f_\mathcal{G}(\mathcal{G}) \tag{9}$$

where $\text{aggr}_\mathcal{G}$ is a node aggregation operation such as $\text{sum}$, $\text{max}$ or $\text{mean}$. The optimization of $q_\phi$ can then be performed by optimizing the MLP encoder $f_{M,\phi_1} : \mathbb{R}^{n+m+r} \to \mathbb{R}^d$, the GNN encoder $f_{\mathcal{G},\phi_2} : \mathbb{R}^{n \times v} \to \mathbb{R}^{n \times d_\mathcal{G}}$ and the encoding aggregator $q_{H,\phi_3} : \mathbb{R}^{d+d_\mathcal{G}} \to ([0,1]^d)^\Sigma$ where $\phi = (\phi_1, \phi_2, \phi_3)$. We designed such construction so that $Z$ possesses a node-level graph embedding $Z_\mathcal{G}$ that enables more involved decoding techniques than generic MLP vector decoding, while the calculation of term $\text{KL}\left[q_\phi(Z|Y, \mathcal{G}, X, T) \parallel q_\phi(Z|Y', \mathcal{G}, X, T')\right]$ can also be reduced to the KL-divergence between the conditional distributions of two graph-level vector representations $Z_H$ and

$Z'_H$ (since $Z_G$ is deterministic and invariant across subjects). In decoding, we construct

$$Y = \left( \bigg\|_{i=1}^{n} f_H \left( Z_{G(i)}, Y_M \right) \right)^{\top} \cdot Y_M \tag{10}$$

$$Y_M \sim p_M(Z_H, T) \tag{11}$$

where $\|$ represents vector concatenation and optimize $p_\theta$ by optimizing the MLP decoder $p_{M,\theta_1} : \mathbb{R}^{d+r} \to ([0,1]^d)^\Sigma$ and the decoding aggregator $f_{H,\theta_2} : \mathbb{R}^{d+d_G} \to \mathbb{R}^d$ where $\theta = (\theta_1, \theta_2)$. The decoding aggregator maps graph embedding $Z_{G(i)}$ of the $i$-th node along with feature vector $Y_M$ to outcome $Y_{(i)}$ of the $i$-th dimension, for which we use an attention mechanism and let $Y_{(i)} = \text{att}_{\theta_2}(Z_{G(i)}, Y_M)^{\top} \cdot Y_M$ where $\text{att}(Q_{(i)}, K)$ gives the attention score for each of $K$'s feature dimensions given a querying node embedding vector $Q_{(i)}$ (i.e., a row vector of $Q$). One can simply use a key-independent attention mechanism $Y_{(i)} = \text{att}_{\theta_2}(Z_{G(i)})^{\top} \cdot Y_M$ if GPU memory is of concern. See Figure 3 for a visualization of the encoding and decoding models. A complexity analysis of the model can be found in Appendix D.

## 2.3 RELATION GRAPH REFINEMENT

Since GRNs are often highly context-specific (Oliver, 2000; Romero et al., 2012), and experimental methods such as ATAC-seq and Chip-seq are prone to false positives, we provide an option to impute and refine a prior GRN by learning from the expression dataset of interest. We propose an approach to update the adjacency matrix in GCN training while maintaining sparse graph operations, and use it in an auto-regressive pre-training task to obtain an updated GRN.

Let $g(\cdot)$ be a GCN with learnable edge weights between all nodes. We aim to acquire an updated adjacency matrix by thresholding updated edge weights post optimization. In practice, such GCNs with complete graphs performed poorly on our task and had scalability issues. Hence we apply dropouts to edges in favor of the initial graph — edges present in $\tilde{\mathcal{E}} = |\mathcal{E} - I| + I$ ($I$ is the identity matrix) are accompanied with a low dropout rate $r_l$ and edges not present in $\tilde{\mathcal{E}}$ with a high dropout rate $r_h$ ($r_l \ll r_h$). A graph convolutional layer of $g(\cdot)$ is then given as

$$H^{l+1} = \sigma(\text{softmax}_r(M \odot L)H^l \Theta^l) \tag{12}$$

where $L \in \mathbb{R}^{n \times n}$ is a dense matrix containing logits of the edge weights and $M \in \mathbb{R}^{n \times n}$ is a sparse mask matrix where each element $M_{i,j}$ is sampled from $Bern(I(\tilde{\mathcal{E}}_{i,j} = 1)r_l + I(\tilde{\mathcal{E}}_{i,j} = 0)r_h)$ in every iteration; $\odot$ is element-wise matrix multiplication; $\text{softmax}_r$ is row-wise softmax operation; $H^l \in \mathbb{R}^{n \times d_l}$, $H^{l+1} \in \mathbb{R}^{n \times d_{l+1}}$ are the latent representations of $\mathcal{V}$ after the $l$-th, $(l + 1)$-th layer; $\Theta^l \in \mathbb{R}^{d_l \times d_{l+1}}$ is the weight matrix of the $(l + 1)$-th layer; $\sigma$ is a non-linear function. The updated adjacency matrix $\hat{\mathcal{E}} \in \mathbb{R}^{n \times n}$ is acquired by rescaling and thresholding the unnormalized weight matrix $W = \exp(L)$ after optimizing $g(\cdot)$:

$$\hat{\mathcal{E}}_{i,j} = \text{sgn} \left| (1 + W_{i,j}^{-1})^{-1} - \alpha \right| \tag{13}$$

where $\alpha$ is a threshold level. We define $\tilde{W} \in \mathbb{R}^{n \times n}$ to be the rescaled weight matrix having $\tilde{W}_{i,j} = (1 + W_{i,j}^{-1})^{-1}$. With this design, the convolution layer operates on sparse graphs which benefit performance and scalability, while each absent edge in the initial graph still has an opportunity to come into existence in the updated graph.

We use this approach to obtain an updated GRN $\hat{\mathcal{E}}$ prior to main model training presented in the previous sections. We train $g(\cdot) : \mathbb{R}^{n \times (1+v+m)} \to \mathbb{R}^{n \times 1}$ on a simple node-level prediction task where the output of the $i$-th node is the expression $Y_{(i)}$ of the $i$-th dimension; the input of the $i$-th node is a combination $O'_{(i)} = (Y_{(i)}, \mathcal{V}_{(i)}, X)$ with expression $Y_{(i)}$, gene features $\mathcal{V}_{(i)}$ of the $i$-th gene and cell covariates $X$. Essentially, we require the model to predict the expression of a gene (node) from its neighbors in the graph. This task is an effective way to learn potential connections in a gene regulatory network as regulatory genes should be predictive of their targets (Kamimoto et al., 2020). The objective is a lasso-style combination of reconstruction loss and edge penalty:

$$J(g) = -\|g(O) - Y\|_{L^2} - \omega \cdot \|\tilde{W}\|_{L^1} \tag{14}$$

where $\omega$ is a scaling coefficient. Note that although $\mathcal{V}_{(i)}$ is not cell-specific and $X$ is not gene-specific, the combination of $(\mathcal{V}_{(i)}, X)$ forms a unique representation for each gene of each cell

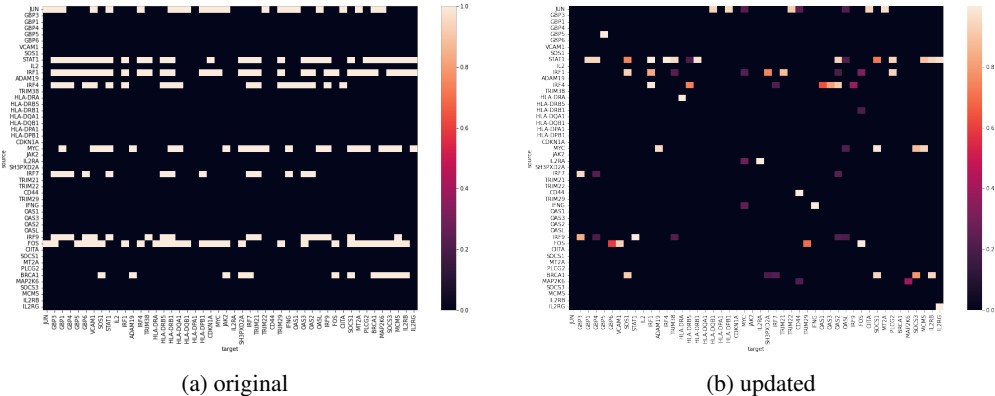

(a) original            (b) updated

Figure 4: An example of an updated gene regulatory network $U$ (where $U_{i,j} = |\tilde{W}_{i,j} - \alpha|$ with $\alpha = 0.2$) after refining the original ATAC-seq-based network Kamimoto et al. (2020) using the Schmidt et al. (2022) dataset. Source nodes are shown as rows and targets are shown as columns for key immune-related genes. The learned edge weights in (b) recapitulate known biology such as STAT1 regulating IRF1. Also note that while many of the edges are present in the original ATAC-seq data from (a), we see some novel edges in (b) such as IFNg regulating MYC (Ramana et al., 2000).

type. We employ such dummy task since node-level predictions with strong self-connections grants interpretability, but additional penalization on the diagonal of $\tilde{W}$ can be applied if one wishes to weaken the self-connections. Otherwise, although self-connections are the strongest signals in this task, dropouts and data correlations will still prevent them from being the only notable signals. See Figure 4 for an example of an updated GRN in practice.

## 3 EXPERIMENTS

We tested our framework on three datasets in experiments. We employ the publicly available sci-Plex dataset from Srivatsan et al. (2020) (**Sciplex**) and CRISPRa dataset from Schmidt et al. (2022) (**Marson**). Sci-Plex is a method for pooled screening that relies on nuclear hashing, and the Sciplex dataset consists of three cancer cell lines (A549, MCF7, K562) treated with 188 compounds. Marson contains perturbations of 73 unique genes where the intervention served to increase the expression of those genes. In addition, we open source in this work a new dataset (**L008**) designed to showcase the power of our model in conjunction with modern genomics.

**L008 dataset**     We used the CROP-seq platform (Shifrut et al., 2018) to knock out 77 genes related to the interferon gamma signaling pathway in CD4$^+$ T cells. They include genes at multiple steps of the interferon gamma signaling pathway such as JAK1, JAK2 and STAT1. We hope that by including multiple such genes, machine learning models will learn the signaling pathway in more detail.

**Baseline**     We compare our framework to three state-of-the-art self-supervised models for individual counterfactual outcome generation — **CEVAE** (Louizos et al., 2017), **GANITE** (Yoon et al., 2018) and **CPA** (Lotfollahi et al., 2021b), along with the non-graph version of our framework **VCI** (Wu et al., 2022b). To give an idea how well these models are doing, we also compare them to a generic autoencoder (**AE**) with covariates and treatment as additional inputs, which serves as an ablation study for all other baseline models. For this generic approach, we simply plug in counterfactual treatments instead of factual treatments during test time.

### 3.1 OUT-OF-DISTRIBUTION PREDICTIONS

We evaluate our model and benchmarks on a widely accepted and biologically meaningful metric — the $R^2$ (coefficient of determination) of the average prediction against the true average from the out-of-distribution (OOD) set (see Appendix E.2) on all genes and differentially-expressed (DE) genes (see Appendix E.1). Same as Lotfollahi et al. (2021a), we calculate the $R^2$ for each perturbation

of each covariate level (e.g. each cell type of each donor), then take the average and denote it as $\bar{R}^2$. Table 1 shows the mean and standard deviation of $\bar{R}^2$ for each model over 5 independent runs. Training setups can be found in Appendix E.3.

Table 1: $\bar{R}^2$ of OOD predictions

| | Sciplex | | Marson | | L008 | |
|---|---|---|---|---|---|---|
| | all genes | DE genes | all genes | DE genes | all genes | DE genes |
| AE[§] | $0.740 \pm 0.043$ | $0.421 \pm 0.021$ | $0.804 \pm 0.020$ | $0.448 \pm 0.009$ | $0.948 \pm 0.010$ | $0.729 \pm 0.041$ |
| CEVAE | $0.760 \pm 0.019$ | $0.436 \pm 0.014$ | $0.795 \pm 0.014$ | $0.424 \pm 0.015$ | $0.941 \pm 0.010$ | $0.632 \pm 0.034$ |
| GANITE[¶] | $0.751 \pm 0.013$ | $0.417 \pm 0.014$ | $0.795 \pm 0.017$ | $0.443 \pm 0.025$ | $0.946 \pm 0.009$ | $0.730 \pm 0.030$ |
| CPA | $0.836 \pm 0.002$ | $0.474 \pm 0.014$ | $0.876 \pm 0.005$ | $0.549 \pm 0.019$ | $0.962 \pm 0.005$ | $0.849 \pm 0.021$ |
| VCI | $0.828 \pm 0.006$ | $0.492 \pm 0.011$ | $0.884 \pm 0.010$ | $0.604 \pm 0.044$ | $0.962 \pm 0.002$ | $\mathbf{0.865} \pm 0.038$ |
| graphVCI | $\mathbf{0.841} \pm 0.002$ | $\mathbf{0.497} \pm 0.014$ | $\mathbf{0.892} \pm 0.003$ | $\mathbf{0.642} \pm 0.016$ | $\mathbf{0.965} \pm 0.002$ | $0.831 \pm 0.018$ |

As can be seen from these experiments, our variational Bayesian causal inference framework with refined relation graph achieved a significant advantage over other models on all genes of the OOD set, and a remarkable advantage on DE genes on Marson. Note that losses were evaluated on all genes during training and DE genes were not being specifically optimized in these runs.

We also examined the predicted distribution of gene expression for various genes and compared to experimental results. Fig 5 shows an analysis of the CRISPRa dataset where MAP4K1 and GATA3 were overexpressed in $CD8^+$ T cells (Schmidt et al., 2022), but these cells were not included in the model's training set. Nevertheless, the model's prediction for the distribution of gene expression frequently matches the ground truth. Quantitative agreement can be obtained from Table 1.

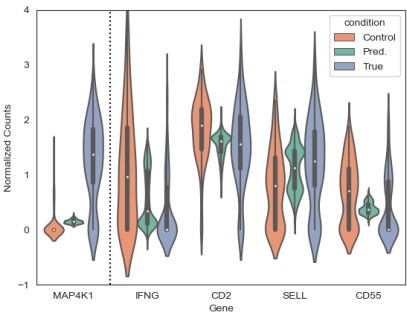
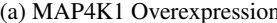
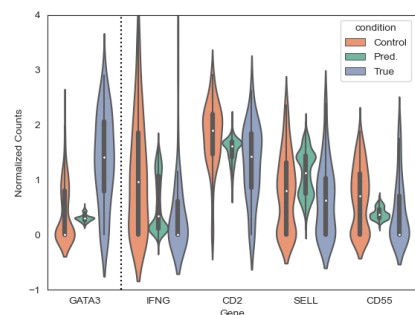

(a) MAP4K1 Overexpression        (b) GATA3 Overexpression

Figure 5: Model predictions versus true distributions for overexpression of genes in CRISPRa experiments (Schmidt et al., 2022). For two perturbations in $CD8^+$ T cells, (a) MAP4K1 overexpression and (b) GATA3 overexpression, we plot the distribution of gene expressions for unperturbed cells ("Control"), the model's prediction of perturbed gene expressions using unperturbed cells as factual inputs ("Pred"), and the true gene expressions for perturbed cells ("True"). The predicted distributional shift relative to control often matches the direction of the true shift.

For graphVCI, we used the key-dependent attention (see Section 2.2) for the decoding aggregator in all runs and there were a few interesting observations we found in these experiments. Firstly, the key-independent attention is more prone to the quality of the GRN and exhibited a more significant difference on model performance with the graph refinement technique compared to the key-dependent attention. Secondly, with graphVCI and the key-dependent attention, we are able to get stable performances across runs while setting $\omega_1$ to be much higher than that of VCI.

---

[§]Autoencoder with covariates and treatment as additional inputs.

[¶]GANITE's counterfactual block. GANITE's counterfactual generator does not scale with a combination of high-dimensional outcome and multi-level treatment, hence we made the same adaptation as Wu et al. (2022b).

## 3.2 Graph Evaluations

In this section, we perform some ablation studies and analysis to validate the claim that the adjacency updating procedure improves the quality of the GRN. We first examine the performance impact of the refined GRN over the original GRN derived from ATAC-seq data. For this purpose, we used the key-independent attention decoding aggregator and conducted 5 separate runs with the original GRN and refined GRN on the Marson dataset (Fig 6a). We found that the refined GRN helps the graphVCI to learn faster and achieve better predictive performance of gene expression, suggesting the updated graph contains more relevant gene connections for the model to predict counterfactual expressions. Note that there is a difference in length of the curves and bands in Fig 6a because we applied early stopping criteria to runs similar to CPA.

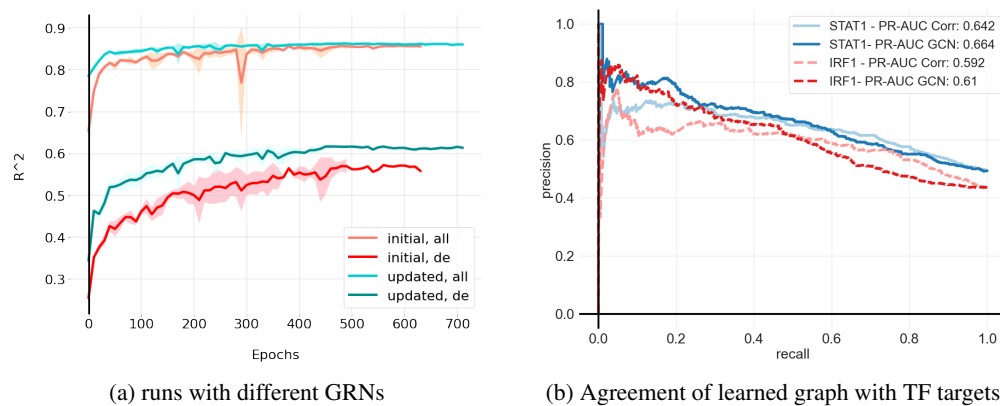

(a) runs with different GRNs

(b) Agreement of learned graph with TF targets

Figure 6: (a) Graph refinement improves model training. We learned a GRN with graph refinement as described in section 2.3, and edges were retained using a threshold of $\alpha = 0.3$. Following the refinement, the model is better able to reconstruct all genes and differentially expressed (DE) genes during training as can be seen in a graph of $R^2$ vs. number of training epochs. (b) We examine whether the edges learned in refinement are accurate by comparing to a database of targets for two important TFs, STAT1 and IRF1. Refined edges between these TFs and their targets agree well with the interactions in the database, according to a precision-recall curve where we treat the database as the true label and edge weights as a probability of interaction. We also compare the refined edges ("GCN") to a gene-gene correlation benchmark ("Corr") and find that the refined graph can outperform the benchmark.

Next, we compared the learned edge weights from our method to known genetic interactions from the ChEA transcription factor targets database (Lachmann et al., 2010). We treat edge weights following graph refinement as a probability of an interaction and treat the known interactions in the database as ground truth for two key TFs, STAT1 and IRF1. We found the refined GRN obtained by graph refinement is able to place higher weights on known targets than a more naive version of the GRN based solely on gene-gene correlation in the same dataset (Fig 6b). The improvement is particularly noticeable in the high-precision regime where we expect the ground truth data is more accurate, since it is expected that a database based on ChIP-seq would contain false positives.

## 4 Conclusion

In this paper, we developed a theoretically grounded novel model architecture to combine deep graph representation learning with variational Bayesian causal inference for the prediction of single-cell perturbation effect. We proposed an adjacency matrix updating technique producing refined relation graphs prior to model training that was able to discover more relevant relations to the data and enhance model performances. Experimental results showcased the advantage of our framework compared to state-of-the-art methods. In addition, we included ablation studies and biological analysis to generate sensible insights. Further studies could be conducted regarding complete out-of-distribution prediction — the prediction of treatment effect when the treatment is completely held out, by rigorously incorporating treatment relations into our framework on top of outcome relations.

ACKNOWLEDGMENTS

We thank Balasubramaniam Srinivasan, Drausin Wulsin, Meena Subramaniam, and Maxime Dhainaut for the insightful discussions. Work by author Robert A. Barton was done prior to joining Amazon.

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

## APPENDIX

## A  LIST OF NOTATIONS

| | |
|---|---|
| $Y$ | outcome random vector / gene expressions |
| $X$ | covariates random vector (or variable) / cell type, donor, replicate, etc. |
| $Z$ | latent features random vector |
| $T$ | treatment random vector (or variable) / perturbation |
| $V'$ | counterfactual random vector of a random vector $V$ |
| $D$ | concatenation of all vectors (and variables) above |
| $n$ | outcome dimensions / number of genes |
| $m$ | combined dimensions of covariates |
| $r$ | treatment dimensions / number of perturbations |
| $\Sigma$ | event space |
| $P$ | probability measure on $\Sigma$ |
| $E_V$ | measurable space generated by a random vector (or variable) $V$ on $\Omega$ |
| $c$ | a covariate value / a sample in $E_X$ |
| $a$ | a treatment value / a sample in $E_T$ |
| $\mathcal{G}$ | relation graph / gene graph |
| $\mathcal{E}$ | adjacency matrix / gene relation network |
| $\mathcal{V}$ | feature matrix / gene feature matrix |
| $\mathcal{D}$ | concatenation of $D$ and $\mathcal{G}$ (see $\mathcal{G}$ as a fixed random vector with its components flattened) |
| $v$ | feature dimensions of $\mathcal{G}$ / diagonal length of $\mathcal{E}$ / column dimensions of $\mathcal{V}$ |
| $A_{(i)}$ | the $i$-th row of a matrix $A$ |
| $A_{i,j}$ | the element on the $i$-th row and $j$-th column of a matrix $A$ |
| $J$ | objective function |
| $KL$ | Kullback–Leibler divergence |
| $\omega.$ | scaling coefficient |
| $p_\theta$ | estimating model for $p(Y|Z,T)$ (and $p(Y'|Z,T')$) |
| $q_\phi$ | estimating model for $q(Z|Y,\mathcal{G},X,T)$ |
| $\tilde{Y}'_{\theta,\phi}$ | a sample from $E_{q_\phi(Z|Y,\mathcal{G},X,T)}p_\theta(Y'|Z,T')$ |
| $\tilde{Z}'_\phi$ | a sample from $q_\phi(Z|Y,\mathcal{G},X,T)$ |
| $p_M$ | true model for the distribution of latent $Y_M$ |
| $q_H$ | true model for the distribution of latent $Z_H$ |
| $f_H$ | true model for the attention score on $Y_M$ |
| $f_M$ | true model for latent $Z_M$ |
| $f_{\mathcal{G}}$ | true model for latent $Z_{\mathcal{G}}$ |
| $h_{a,\beta}$ | estimating model for a true model $h_a$ with a parameterization $\beta$ |
| $d$ | dimensions of $Z_M, Z_H, Y_M$ |
| $d_{\mathcal{G}}$ | feature dimensions of $Z_{\mathcal{G}}$ |
| att | alias for $f_H$ |
| $\text{aggr}_{\mathcal{G}}$ | a graph node aggregation operation / a matrix row aggregation operation |
| $\text{softmax}_r$ | row-wise softmax function |
| $\odot$ | element-wise matrix multiplication |
| $\sigma$ | a non-linear function |
| $g$ | graph convolution network |
| $r.$ | a probability value |
| $\alpha$ | a threshold value for probability values |
| $I$ | identity matrix |
| $M$ | mask matrix / probability matrix |
| $L$ | edge weight logits matrix |
| $W$ | unnormalized edge weight matrix |
| $\tilde{W}$ | rescaled weight matrix / matrix after applying element-wise sigmoid function on $L$ |
| $U$ | matrix after thresholding $\tilde{W}$ |
| $O$ | input node feature matrix to $g$ in the graph refinement task |
| $H^l$ | latent representation matrix after the $l$-th layer of $g$ |
| $\Theta^l$ | weight matrix of the $l$-th layer of $g$ |
| $\Psi$ | causal parameter |

## B    PROOF OF THEOREM 1

*Proof.* By the d-separation (Pearl, 1988) of paths on the causal graph defined in Figure 1, we have

$$\log\left[p(Y'|Y,\mathcal{G},X,T,T')\right] = \log\mathbb{E}_{p(Z|Y,\mathcal{G},X,T)}\left[p(Y'|Z,Y,\mathcal{G},X,T,T')\right] \tag{15}$$

$$\geq \mathbb{E}_{p(Z|Y,\mathcal{G},X,T)}\log\left[p(Y'|Z,Y,\mathcal{G},X,T,T')\right] \quad \text{(Jensen's ineq.)} \tag{16}$$

$$= \mathbb{E}_{p(Z|Y,\mathcal{G},X,T)}\log\frac{p(Y',Z|Y,\mathcal{G},X,T,T')}{p(Z|Y,\mathcal{G},X,T)} \tag{17}$$

$$= \mathbb{E}_{p(Z|Y,\mathcal{G},X,T)}\log\frac{p(Y',Z,Y,\mathcal{G},T|X,T')}{p(Z|Y,\mathcal{G},X,T)p(Y,T|\mathcal{G},X)p(\mathcal{G})} \tag{18}$$

$$= \mathbb{E}_{p(Z|Y,\mathcal{G},X,T)}\log\frac{p(Y'|\mathcal{G},X,T')p(Z|Y',\mathcal{G},X,T')p(Y,T|Z,X)}{p(Z|Y,\mathcal{G},X,T)}$$
$$- \log\left[p(Y,T|\mathcal{G},X)\right] \tag{19}$$

$$= \log\left[p(Y'|\mathcal{G},X,T')\right] - \text{KL}\left[p(Z|Y,\mathcal{G},X,T) \parallel p(Z|Y',\mathcal{G},X,T')\right]$$
$$+ \mathbb{E}_{p(Z|Y,\mathcal{G},X,T)}\log\left[p(Y|Z,T)p(T|X)\right] - \log\left[p(Y,T|\mathcal{G},X)\right] \tag{20}$$

$$= \log\left[p(Y'|\mathcal{G},X,T')\right] - \text{KL}\left[p(Z|Y,\mathcal{G},X,T) \parallel p(Z|Y',\mathcal{G},X,T')\right]$$
$$+ \mathbb{E}_{p(Z|\mathcal{G},Y,X,T)}\log\left[p(Y|Z,T)\right] - \log\left[p(Y|\mathcal{G},X,T)\right] \tag{21}$$

Reorganizing the terms yields

$$\log\left[p(Y'|Y,\mathcal{G},X,T,T')\right] + \log\left[p(Y|\mathcal{G},X,T)\right] \geq \mathbb{E}_{p(Z|Y,\mathcal{G},X,T)}\log\left[p(Y|Z,T)\right]$$
$$+ \log\left[p(Y'|\mathcal{G},X,T')\right] - \text{KL}\left[p(Z|Y,\mathcal{G},X,T) \parallel p(Z|Y',\mathcal{G},X,T')\right] \tag{22}$$

$\square$

## C    MARGINAL EFFECT ESTIMATION

### C.1    EXPERIMENT

To evaluate the marginal estimator $\hat{\Psi}_{\theta,\phi}$ in Equation 5, we compute $\hat{\Psi}_{\theta,\phi}$ for treatment $a$ and co-variate level $c$ with samples from the training set and calculate its $R^2$ against the true average of the samples with treatment $a$ and covariate level $c$ in the validation set. We record the average $R^2$ of all treatment-covariate combo similar to Section 3.1, and compare it (robust) to that of the regular empirical mean estimator (mean). Table 2 shows the results on Marson (Schmidt et al., 2022) episodically during training.

Table 2: Comparison of marginal estimators on Marson

| Episode | All Genes | | DE Genes | |
|---|---|---|---|---|
| | mean | robust | mean | robust |
| 40 | $0.9177 \pm 0.0015$ | $\mathbf{0.9329 \pm 0.0008}$ | $0.7211 \pm 0.0116$ | $\mathbf{0.9048 \pm 0.0040}$ |
| 80 | $0.9193 \pm 0.0019$ | $\mathbf{0.9337 \pm 0.0008}$ | $0.7339 \pm 0.0149$ | $\mathbf{0.9076 \pm 0.0043}$ |
| 120 | $0.9178 \pm 0.0037$ | $\mathbf{0.9340 \pm 0.0008}$ | $0.7234 \pm 0.0175$ | $\mathbf{0.9104 \pm 0.0038}$ |
| 160 | $0.9191 \pm \mathbf{0.0009}$ | $\mathbf{0.9356} \pm 0.0010$ | $0.7343 \pm 0.0079$ | $\mathbf{0.9175 \pm 0.0034}$ |

These runs reflects that the robust estimator was able to produce a more accurate estimation of the covariate-stratified marginal treatment effect $\mathbb{E}_p(Y'|X=c,T'=a)$ with a tigher confidence bound.

### C.2    DERIVATION

By Van der Vaart (2000), we derive the efficient influence function of $\Psi(p)$ and thus provides a mean for asymptotically efficient estimation:

**Theorem 2.** *Suppose $\mathcal{D}:\Omega \to E$ follows a causal structure defined by the Bayesian network in Figure 1, where the counterfactual conditional distribution $p(Y',T'|Z,X)$ is identical to that of its factual counterpart $p(Y,T|Z,X)$. Then $\Psi(p)$ has the following efficient influence function:*

$$\tilde{\psi}(p) = \frac{I(X=c,T=a)}{p(X,T)}(Y - \mathbb{E}_p\left[Y|Z,T\right]) + \frac{I(X=c)}{p(X)}(\mathbb{E}_p\left[Y'|Z,T'=a\right] - \Psi). \tag{23}$$

*Proof.* Following Van der Vaart (2000), we define a path $p_\epsilon(\mathcal{D}) = p(\mathcal{D})(1 + \epsilon S(\mathcal{D}))$ on density $p$ of $\mathcal{D}$ as a submodel that passes through $p$ at $\epsilon = 0$ in the direction of the score $S(\mathcal{D}) = \frac{d}{d\epsilon} \log [p_\epsilon(\mathcal{D})]\Big|_{\epsilon=0}$. Let $\mathcal{X} = (\mathcal{G}, X)$ and minuscule of a variable denote the value it takes. By Levy (2019), we have

$$\frac{d}{d\epsilon}\Psi(p_\epsilon)\Big|_{\epsilon=0} = \frac{d}{d\epsilon}\Big|_{\epsilon=0} \mathbb{E}_{p_\epsilon}\left[\mathbb{E}_{p_\epsilon}\left[Y'|Z, T' = a\right]|X = c\right] \tag{24}$$

$$= \frac{d}{d\epsilon}\Big|_{\epsilon=0} \int_{y',z} y'\left[p_\epsilon(y'|z, T' = a)p_\epsilon(z|X = c)\right] \tag{25}$$

$$= \int_{y',z} y' \frac{d}{d\epsilon}\Big|_{\epsilon=0}\left[p_\epsilon(y'|z, T' = a)p_\epsilon(z|X = c)\right] \quad \text{(dominated convergence)} \tag{26}$$

$$= \int_{y',z} y' p(z|X = c)\frac{d}{d\epsilon}\Big|_{\epsilon=0} p_\epsilon(y'|z, T' = a) \tag{27}$$

$$+ \int_{y',z} y' p(y'|z, T' = a)\frac{d}{d\epsilon}\Big|_{\epsilon=0} p_\epsilon(z|X = c) \tag{28}$$

$$= \int_{\mathcal{d}} I(x = c, t' = a)\frac{p(\mathcal{x}, t')}{p(x, t')} y' p(y, t|z, x) p(z|\mathcal{x})\frac{d}{d\epsilon}\Big|_{\epsilon=0} p_\epsilon(y'|z, t')$$
$$+ \int_{y',z,\mathcal{x}} I(x = c)\frac{p(\mathcal{x})}{p(x)} y' p(y'|z, T' = a)\frac{d}{d\epsilon}\Big|_{\epsilon=0} p_\epsilon(z|\mathcal{x}) \tag{29}$$

$$= \int_{\mathcal{d}} \frac{I(x = c, t' = a)}{p(x, t')} y' p(y, y', t, t'|z, x) p(z, \mathcal{x})\left\{S(\mathcal{d}) - \mathbb{E}\left[S(\mathcal{D})|y, z, \mathcal{x}, t, t'\right]\right\}$$
$$+ \int_{y',z,\mathcal{x}} \frac{I(x = c)}{p(x)} y' p(y'|z, T' = a) p(z, \mathcal{x})\left\{\mathbb{E}\left[S(\mathcal{D})|z, \mathcal{x}\right] - \mathbb{E}\left[S(\mathcal{D})|\mathcal{x}\right]\right\} \tag{30}$$

$$= \int_{\mathcal{d}} \frac{I(x = c, t = a)}{p(x, t)} y p(y', y, t', t|z, x) p(z, \mathcal{x})\left\{S(\mathcal{d}) - \mathbb{E}\left[S(\mathcal{D})|y', z, \mathcal{x}, t', t\right]\right\}$$
$$+ \int_{y',z,\mathcal{x}} \frac{I(x = c)}{p(x)} y' p(y'|z, T' = a) p(z, \mathcal{x})\left\{\mathbb{E}\left[S(\mathcal{D})|z, \mathcal{x}\right] - \mathbb{E}\left[S(\mathcal{D})|\mathcal{x}\right]\right\} \tag{31}$$

$$= \int_{\mathcal{d}} S(\mathcal{d}) \cdot \frac{I(x = c, t = a)}{p(x, t)} y p(\mathcal{d})$$
$$- \int_{\mathcal{d}} \mathbb{E}\left[S(\mathcal{D})|y', z, \mathcal{x}, t, t'\right] p(y', z, \mathcal{x}, t, t') \cdot \frac{I(x = c, t = a)}{p(x, t)} y p(y|z, t)$$
$$+ \int_{y',z,\mathcal{x}} \mathbb{E}\left[S(\mathcal{D})|z, \mathcal{x}\right] p(z, \mathcal{x}) \cdot \frac{I(x = c)}{p(x)} y' p(y'|z, T' = a)$$
$$- \int_{y',z,\mathcal{x}} \mathbb{E}\left[S(\mathcal{D})|\mathcal{x}\right] \cdot \frac{I(x = c)}{p(x)} y' p(y'|z, T' = a) p(z, \mathcal{x}) \tag{32}$$

$$= \int_{\mathcal{d}} S(\mathcal{d})\left\{\frac{I(x = c, t = a)}{p(x, t)}(y - \mathbb{E}\left[Y|z, t\right])\right.$$
$$\left. + \frac{I(x = c)}{p(x)}(\mathbb{E}\left[Y'|z, T' = a\right] - \Psi)\right\} p(\mathcal{d}) \tag{33}$$

by assumptions of Theorem 2 and factorization according to Figure 1. Hence

$$\frac{d}{d\epsilon}\Psi(p_\epsilon)\Big|_{\epsilon=0} = \left\langle S(\mathcal{D}), \frac{I(X = c, T = a)}{p(X, T)}(Y - \mathbb{E}_p\left[Y|Z, T\right])\right.$$
$$\left. + \frac{I(X = c)}{p(X)}(\mathbb{E}_p\left[Y'|Z, T' = a\right] - \Psi)\right\rangle_{L^2(\Omega;E)} \tag{34}$$

and we have $\tilde{\psi}_p = \frac{I(X=c,T=a)}{p(X,T)}(Y - \mathbb{E}_p\left[Y|Z, T\right]) + \frac{I(X=c)}{p(X)}(\mathbb{E}_p\left[Y'|Z, T' = a\right] - \Psi)$. $\qquad\square$

By Theorem 2, we propose the following estimator that is asymptotically efficient among regular estimators under some regularity conditions (Van Der Laan & Rubin, 2006):

$$\hat{\Psi}_{\theta,\phi} = \frac{1}{n} \sum_{k=1}^{n} \left\{ \frac{I(T_k = a, X_k = c)}{\hat{p}(T_k|X_k)\hat{p}(X_k)} \left[ Y_k - \mathbb{E}_{p_\theta}(Y'|\tilde{Z}_{k,\phi}, T'_k = a) \right] \right.$$
$$\left. + \frac{I(X_k = c)}{\hat{p}(X_k)} \mathbb{E}_{p_\theta}(Y'|\tilde{Z}_{k,\phi}, T'_k = a) \right\} \tag{35}$$

where $(Y_k, X_k, T_k)$ are the observed variables of the $k$-th individual and $\tilde{Z}_{k,\phi} \sim q_\phi(Y_k, \mathcal{G}, X_k, T_k)$; $\hat{p}(T|X)$ is an estimation of the propensity score and $\hat{p}(X)$ is an estimation of the density of $X$. In the context of this work, $X$ and $T$ are discrete, hence $\hat{p}(T|X)$ and $\hat{p}(X)$ can be estimated by the empirical density $p_n(T|X)$ and $p_n(X)$. The above estimator then reduces to:

$$\hat{\Psi}_{\theta,\phi} = \frac{1}{n_{a,c}} \sum_{k=1_{a,c}}^{n_{a,c}} \left\{ Y_k - \mathbb{E}_{p_\theta}(Y'|\tilde{Z}_{k,\phi}, T'_k = a) \right\}$$
$$+ \frac{1}{n_c} \sum_{k=1_c}^{n_c} \left\{ \mathbb{E}_{p_\theta}(Y'|\tilde{Z}_{k,\phi}, T'_k = a) \right\} \tag{36}$$

where $(1_c, \ldots, n_c)$ are the indices of the observations having $X = c$ and $(1_{a,c}, \ldots, n_{a,c})$ are the indices of the observations having both $T = a$ and $X = c$.

## D  COMPLEXITY ANALYSIS

Overall, the time complexity of VCI compared to a generic framework like VAE (or the time complexity of graphVCI compared to a generic GNN framework like GLUE (Cao & Gao, 2022) does not over-scale on any factor of any parameter – to put it simply, the workflow of VCI is just twice the forward passes of an VAE for a batch of inputs, with an additional distribution loss which is implemented on the same scale $O(r)$ as other losses. Comparing graphVCI to VCI, graphVCI has additionally a few graph operations. We give a thorough analysis of the time complexity of each layer in our experiments below:

- Generic method AE has 2 MLP layers with $O(rd)$ number of operations and 4 layers of $O(d^2)$ number of operations. $d$ is the number of hidden neurons.
- VCI has the same layer sizes: 2 MLP layers with $O(rd)$ number of operations and 4 layers of $O(d^2)$ number of operations. But every layer is forward passed twice.
- graphVCI has 1 GNN layer with $O(rvd_g^2 + pd_g r^2)$ number of operations. $v$ is number of gene features, $d_g$ is number of hidden neurons for GNN (usually a lot less than $d$ since $v \ll r$) and $p$ is the sparsity of GRN (number of connections divided by $r^2$, usually around 1%), and the following layers which are forward passed twice: 1 MLP layer and 1 dot-product operation each with $O(rd)$ number of operations, 1 MLP layer with $O(rd_g(d_g+d))$ number of operations and 3 MLP layers with $O(d^2)$ number of operations.

So the terms to be concerned compared to AE and VCI are $O(rvd_g^2)$, $O(pd_g r^2)$ and $O(rd_g d)$. Since $p$ is around 1% and $r$ is 2000 in our experiment, $pd_g r^2 \approx 20d_g r$ hence the second term is comparably smaller than the other two terms. Therefore, as long as $d_g$ is set to be reasonably small compared to $d$, the graph approach is reasonably scaled compared to pure MLP approaches. We note that this is a limitation of ours and GNN approaches in general: if there is a much high number of genes $r$ than 2000 to be considered, or a high number of gene features $v$ for each gene (which results in that $d_g$ has to be higher), GNN methods does not scale favorably compared to MLP methods.

## E  EXPERIMENTS DETAILS

### E.1  DIFFERENTIALLY-EXPRESSED GENES

To evaluate the predictions on the genes that were substantially affected by the perturbations, we select sets of 50 differentially-expressed genes associated with each perturbation and separately report performance on these genes. The same procedure was carried out by Lotfollahi et al. (2021a).

### E.2 OUT-OF-DISTRIBUTION SELECTIONS

We randomly select a covariate category (e.g. a cell type) and hold out all cells in this category that received one of the twenty perturbations whose effects are the hardest to predict. We use these held-out data to compose the out-of-distribution (OOD) set. We computed the euclidean distance between the pseudobulked gene expression of each perturbation against the rest of the dataset, and selected the top twenty most distant ones as the hardest-to-predict perturbations. This is the same procedure carried out by Lotfollahi et al. (2021a).

### E.3 TRAINING SETUP

The data excluding the OOD set are split into training and validation set with a four-to-one ratio. A few additional cell attributes available in each dataset are selected as cell covariates. For Sciplex, they are cell type and replicate; for Marson, they are cell type, donor and stimulation; for L008, they are cell type and donor. All these covariates are categorical and transformed into discrete indicators before passing to the models. All common hyperparameters of all models (network width, network depth, learning rate, decay rate, etc.) are set to the same as Lotfollahi et al. (2021a). More details regarding hyperparameter settings can be found in our code repository[†].

---

[†] https://github.com/yulun-rayn/graphVCI

