# OpenReview forum: "Predicting Cellular Responses with Variational Causal Inference and Refined Relational Information"
_ICLR.cc/2023/Conference — ICLR 2023 poster_

### Official Review · Reviewer_TW7F · 2022-10-22

**Confidence:** 4
**Clarity, Quality, Novelty And Reproducibility:** See above.
**Correctness:** 3
**Technical Novelty And Significance:** 3
**Empirical Novelty And Significance:** 3
**Recommendation:** 5

**Strength And Weaknesses:**

Strengths: 1. Paper is well written and easy to follow.
                  2. The idea of combining graph representation learning with variational bayesian causal inference is interesting, and the method is sound.

Weaknesses: 1. The performance is not significant compared to other competing methods in Table 1.
                       2. There is no model complexity study.

**Summary Of The Paper:**

This paper proposes a graph variational Bayesian causal inference framework to predict a cell's gene expression under counterfactual perturbations, gene regulatory networks were used to aid the individualized cellular response predictions. A robust estimator for the asymptotically efficient estimation of the marginal perturbation effect was also proposed. Empirical studies showed its superiority over state-of-the-art methods.

**Summary Of The Review:**

Overall, I like the idea of this paper, and it seems theoretically sound. My only concern is the performance compared to other methods is not significant, and it lacks model complexity study, so I don't know whether the performance of the model outweighs its computational cost.

---

> ### Author Response · Authors · 2022-11-15
> **Reply to Reviewer TW7F**
>
> Thanks for the review and comments. We understand that the primary concern of yours is whether the performance of the model outweighs its computational cost. We appreciate the concern and have made clarifications as well as additional content accordingly to improve our paper:
>
> 1. The performance is not significant compared to other competing methods in Table 1.
>
> We agree with the reviewer that the performance improvement of our graphVCI in individual outcome prediction reported in Table 1 does not seem significant in some datasets. However, it did reach a competitive level with SotA methods including CPA and VCI. Additionally, the novel relational graph refinement procedure of graphVCI provides explainability that is absent in competing methods. Model explainability is a particularly significant contribution to this task: our framework highlights important edges in the GRN when making predictions for cellular response, which is important for biological interpretation.
>
> 2. There is no model complexity study.
>
> We recognize the lack of model complexity study and give a complexity study below as well as adding it to Appendix E of the paper. First, we point out that the time complexity of VCI compared to a generic framework like VAE (or the time complexity of graphVCI compared to a generic GNN-VAE framework like GLUE (Cao & Gao, 2022)) does not over-scale on any factor of any parameter – to put it simply, the workflow of VCI is just 2x the forward passes of an VAE for a batch of inputs, with an additional distribution loss which we made sure is implemented on the same scale as other losses ($O(r)$ where $r$ is the number of genes). Comparing graphVCI to VCI, graphVCI has additionally a few graph operations, we give a thorough comparison of the time complexity of each layer in our experiments below:
>
> - Generic method AE in Table 1. has 2 MLP layers with $O(r d)$ number of operations and 4 layers of $O(d^2)$ number of operations. $d$ is the number of hidden neurons.
> - VCI has the same layer sizes: 2 MLP layers with $O(r d)$ number of operations and 4 layers of $O(d^2)$ number of operations. But every layer is forward passed twice.
> - graphVCI has 1 GNN layer with $O(r v d^2_g + p d_g r^2)$ number of operations. $v$ is number of gene features, $d_g$ is number of hidden neurons for GNN (usually a lot less than $d$ since $v \ll r$) and $p$ is the sparsity of GRN (number of connections divided by $r^2$, usually around 1%), and the following layers which are forward passed twice: 1 MLP layer and 1 dot-product operation each with $O(r d)$ number of operations, 1 MLP layer with $O(r d_g (d_g +d))$ number of operations and 3 MLP layers with $O(d^2)$ number of operations.
>
> So the terms to be concerned compared to AE and VCI are $O(r v d^2_g)$, $O(p d_g r^2)$ and $O(r d_g d)$. Since $p$ is around 1% and $r$ is 2000 in our experiment, $p d_g r^2 \approx 20 d_g r$ hence the second term is comparably smaller than the other two terms. Therefore, as long as $d_g$ is set to be reasonably small compared to $d$, the graph approach is reasonably scaled compared to pure MLP approaches. We note that this is unavoidable when GNN approaches are considered – the method does not scale favorably compared to MLP-only methods if there is a much high number of genes $r$ than 2000 to be considered, or a high number of gene features $v$ for each gene (which results in that $d_g$ has to be higher). But in the context of the current single-cell experiments, we and a few other works (Cao & Gao, 2022; Roohani et al., 2022) decided that the incorporation of gene graph relations is worth the additional computations.

---

### Official Review · Reviewer_qD32 · 2022-10-23

**Confidence:** 3
**Correctness:** 3
**Technical Novelty And Significance:** 2
**Empirical Novelty And Significance:** 3
**Recommendation:** 6

**Clarity, Quality, Novelty And Reproducibility:**

The clarity and quality of the paper are good. It seems to me that interested readers can reproduce the paper with supplemented codes.


**Strength And Weaknesses:**

Strength: Overall, the paper delivers clear explanations regarding 1. the motivation for cell response under different perturbations, 2. model architectures 3. experimental results discussions. To validate the proposed GraphVCI method, the authors run experiments on several publicly available datasets, which I consider the main experiments are thorough. Besides, the codes are supplemented in the submission, this can help or guide interested readers to reproduce the results.

Weakness: I think the ablation studies such as excluding the counterfactual-related losses and features should be performed in the paper,
this can disentangle the contributions that come from different network components. Second, I don't know why the predicted distributional shift presented in Figure 5 matches the true shift, and I feel that they look quite different from each other. Could the authors explain this to me? In the end, the authors introduced 'the causal relation diagram' in Figure 1. I think it would be important to compare it with the standard causal graph definition such as the structural causal model (SCM).

**Summary Of The Paper:**

In this paper, the authors proposed a Graph Variational Causal Inference (GraphVCI) network to estimate gene response under perturbations.
To this end, the component of (hypothetical) counterfactual perturbations is integrated into the GraphVCI network to assist the reconstruction training of the proposed Graph autoencoder.  On three commonly used datasets and a soon-to-be open-sourced dataset, the proposed model demonstrated clear improvement compared to several baselines.

**Summary Of The Review:**

Overall, I think this is an interesting paper that brings some new insights to the analysis of the cell responses to different perturbations, which can potentially achieve a good impact on the ML for the biology community.

---

> ### Author Response · Authors · 2022-11-15
> **Reply to Reviewer qD32**
>
> We thank you for acknowledging those strengths, we did try to do our best to make the delivery clear and the code readable and reproducible! We also acknowledge the weaknesses you pointed out, and have made a few clarifications:
>
> 1. I think the ablation studies such as excluding the counterfactual-related losses and features should be performed in the paper, this can disentangle the contributions that come from different network components.
>
> This is a very good point and we’d like to note that ablation studies such as excluding the counterfactual-related losses and features are indeed performed in the paper – if you exclude counterfactual-related losses and features, our model is simply AE in table 1. Note that AE also serves as an ablation study for other benchmark approaches in table 1, e.g. CPA without adversarial loss and dose curve is AE. This is the reason why we included AE. We have added this point to the baseline paragraph in section 3 of the paper. Besides, the comparison between VCI and graphVCI and the ablation study performed in section 3.3 also showed the contribution of using graph and the contribution of using graph refinement.
>
> 2. Second, I don't know why the predicted distributional shift presented in Figure 5 matches the true shift, and I feel that they look quite different from each other. Could the authors explain this to me?
>
> Yes you’re right, we meant that the direction of the marginal distribution shift often matches the true shift direction. We can’t actually guarantee that the two distributions would look similar to each other, and we give an explanation as to why this is the case in the paragraph below. We have modified the caption, thanks for bringing this up and sorry for the confusion!
>
> First we want to reiterate that, different from traditional causal inference, one is not able to give individualized supervision (other than self-supervision) for every subject since the outcomes have to be used as inputs. As can be seen from our objective construction, the counterfactual outcome is supervised by (an estimation of) distribution $p(Y’ | X, T’)$ – this intuitively means that we require each point in the predicted distribution (green) to have a reasonably high likelihood of coming from the true distributions (blue). Now given this logic, it won’t be hard to figure out why there is no guarantee that the green distribution as a whole would look like the blue distribution, yet you can generally expect the shift direction to be correct. Note that predictions are made using the control samples as factual inputs, i.e. there is a one-to-one mapping between points in green distribution and points in orange distribution. As discussed, the diversity/variation/individuality comes from the features embedded in the factual outcomes (oranges), hence the variation of the orange distribution is a determining factor of the variation of the counterfactual predictions, i.e. the green distribution. We note that when the true shift isn’t actually available, i.e. there isn’t a one-to-one mapping from the orange distribution to the blue distribution (they are two groups of different subjects), this is always going to be a challenge for the non-traditional causal inference setting. We could conduct further study in the future to see if it’d be a good idea to implement batch-wise supervision – to implement a loss that encourage the empirical distribution of a batch of counterfactual constructions (i.e. a collective group of points in the green distribution) to resemble $p(Y’ | X, T’)$ (i.e. the blue distribution).
>
> 3. In the end, the authors introduced 'the causal relation diagram' in Figure 1. I think it would be important to compare it with the standard causal graph definition such as the structural causal model (SCM).
>
> We agree that this is useful, and we refer to the VCI paper (Wu et al., 2022b) if readers seek a comparison between variational causal inference vs. traditional causal inference or VCI causal diagram vs. traditional SCM. Compared to VCI, graphVCI only added a graph component $\mathcal{G}$ that influences the outcome through latent $Z$.

---

### Official Review · Reviewer_FrCx · 2022-10-23

**Confidence:** 3
**Clarity, Quality, Novelty And Reproducibility:** See above.
**Correctness:** 4
**Technical Novelty And Significance:** 3
**Empirical Novelty And Significance:** 4
**Recommendation:** 8

**Strength And Weaknesses:**

Overall, the paper is strong. The text and figures are understandable. The results have significance both in terms of methodology and application. The experimental results are good.

Novelty is moderate because most of the paper is an extension of (Wu et al 2022b). However, the addition of the GCN to VCI is novel.

There is no description of how the authors go from experimental RNA-seq data to values of X and Y, including in Appendix C, which explains some other parts of the experiments. In particular, I couldn't figure out what X is for their experiments. I think they followed (Lotfollahi et al 2021a) in this, but it was hard to figure out what the authors of this paper did even when examining Lotfollahi et al 2021a. My best guess is that X is a discrete indicator of cell type.

The setup is rather specific to the application to RNA-seq because it requires that each individual has an information-rich outcome. It doesn't seem obvious that the same setup would work in a setting such as clinical drug response, where each individual has information-rich features but simple outcomes (e.g. symptoms resolved or not resolved).

It is unclear whether the revised GCN produced by the method of Sec 2.3 is improved according to the desirable criteria for GCNs. There are many existing methods for producing GCNs, so it would be helpful to argue if this improvement method is a better approach for producing a GCN in general, or if it is a preprocessing step useful only for input to a NN method like this.

**Summary Of The Paper:**

The authors aim to tackle the problem of predicting the effect on gene expression of cellular perturbations such as gene knock-out. The authors build upon Variational Causal Inference (Wu et al 2022b). VCI takes as input a set of features X, given treatment T (e,g, drug treatment or knockout of a particular gene) and outcome Y (RNA-seq following treatment). It aims to predict, for some alternative treatment T', the corresponding outcome Y'. The authors extend VCI by additionally taking as input a gene regulatory network (GCN). The authors evaluate the method by holding out several perturbations for a particular cell type and using their method to predict the held-out values.

**Summary Of The Review:**

See above.

---

> ### Author Response · Authors · 2022-11-15
> **Reply to Reviewer FrCx**
>
> We appreciate the positive and accurate summary! Here we make some clarifications according to your comment:
>
> 1. There is no description of how the authors go from experimental RNA-seq data to values of X and Y, including in Appendix C, which explains some other parts of the experiments. In particular, I couldn't figure out what X is for their experiments. I think they followed (Lotfollahi et al 2021a) in this, but it was hard to figure out what the authors of this paper did even when examining Lotfollahi et al 2021a. My best guess is that X is a discrete indicator of cell type.
>
> Thank you for pointing this out! We should have clarified this in the appendix because it is indeed a little different than Lotfollahi et al 2021a – in their published work, $X$ is just cell type; in ours, $X$ additionally includes all covariates that come with the dataset. For Sciplex, it is cell type and replicate; for Marson, it is cell type, donor and stimulation; for L008, it is cell type and donor. Yes, all these covariates are discrete indicators. We have added this information into Appendix C.3.
>
> 2. The setup is rather specific to the application to RNA-seq because it requires that each individual has an information-rich outcome. It doesn't seem obvious that the same setup would work in a setting such as clinical drug response, where each individual has information-rich features but simple outcomes (e.g. symptoms resolved or not resolved).
>
> Yes this remark is correct! The approach is developed to specifically address high-dimensional outcomes that contain information-rich features. For one-dimensional outcome such as resolved or not resolved, traditional causal machine learning (fitting $p(Y | X, T)$ or $E(Y | X, T)$ model) should be used.
>
> 3. It is unclear whether the revised GCN produced by the method of Sec 2.3 is improved according to the desirable criteria for GCNs. There are many existing methods for producing GCNs, so it would be helpful to argue if this improvement method is a better approach for producing a GCN in general, or if it is a preprocessing step useful only for input to a NN method like this.
>
> We thank the reviewer for highlighting this point. There are indeed many existing methods for producing GRNs and our paper mainly shows that this method is useful as an input to the NN to improve its performance. We have focused on emphasizing the utility of a GRN as an input to our model rather than as an output by itself because (1) ground truth data for GRNs is not always available, whereas the perturbation data is abundant, and (2) it is difficult to capture all of the underlying biological dynamics in a GRN; hence, we feel that an important path forward for the field is to build interpretable models that predict perturbational data.
>
> We also do our best to compare what is learned by the model to a ground truth, which in this case is the ChEA transcription factor targeting DB (Figure 6b). While we show that our results are competitive with a purely correlation-based benchmark, we do not necessarily expect it to outperform all methods of GRN production. The primary value lies in the ability to incorporate interpretable models into a perturbation prediction framework. We have altered the text to make this point clear.

---

> > ### Comment · Reviewer_FrCx · 2022-11-20
> > **2022-11-20**
> >
> > Thank you for the response. My score is unchanged.

---

### Official Review · Reviewer_kcXz · 2022-10-30

**Confidence:** 3
**Correctness:** 3
**Technical Novelty And Significance:** 3
**Empirical Novelty And Significance:** 3
**Recommendation:** 6

**Clarity, Quality, Novelty And Reproducibility:**

The paper is written clearly, though it would benefit from providing even more background on causal inference pertinent to the problem at hand, e.g. differences between individuals and levels. It appears to be both of high-quality and the particular variational formulation is novel.

**Strength And Weaknesses:**

Strengths:
- Novel methods leading to state-of-the-art results compared with other models trained for generating individual counterfactual outcomes
- Paced introduction to help understand the method
- Excellent figures and legends
- Makes use of GRNs in novel way
- Introduces new CROP-seq dataset

Weaknesses:
- Contains a lot of complex notation and indices, some of which is not intuitive (e.g., indices of graph latents)
- It is not clear that the manner in which the GRN is incorporated is necessary, and it is reasonable to expect that a simpler approach could yield equivalent results.
- In Figure 5, the predicted distributions do not often look like the ground-truth distributions. Moreover, many of the ground-truth distributions look like control distributions, and so would be difficult to distinguish from the background anyway.

**Summary Of The Paper:**

The authors propose a novel variational causal inference model for constructing the gene expression counts in cells after counterfactual perturbations that relies on information from two sources: individual features embedded in the outcome Y, and response distributions of similar individuals that did indeed receive the treatment. The method is aided by an additional method for updating the connections of the GRN prior to model training. The authors compare this method to known methods and find that theirs produces state-of-the-art R^2 in out-of-distribution predictions.

**Summary Of The Review:**

The authors do a good job of motivating the variational causal inference formulation, but not the particulars of the graph latents. While some of the results show clear superiority over existing methods, there are overlapping error bars in Table 1, and Figure 5 seems not to show what is claimed. Clarifying these, as well as better motivating why the particular method for generating graph latents, would make this a stronger paper.

---

> ### Author Response · Authors · 2022-11-15
> **Reply to Reviewer kcXz**
>
> We appreciate your summary of our paper and its strengths! Here we address the weaknesses you listed:
>
> 1. Contains a lot of complex notation and indices, some of which is not intuitive (e.g., indices of graph latents)
>
> Thanks for pointing this out! Per your suggestion, we have made some changes regarding the notations, as well as putting a table of notations in Appendix D of the paper.
> - In Section 2.2, we now use an universal notation $d$ for all hidden sizes of MLP.
> - In Section 2.2, we removed all the three-level notations (e.g. $A_{B_C}$).
> - In Section 2.3, notations for element-wise matrix indexing are further clarified and made consistent.
> - Regarding indices of graph latents, we believe using $M_{(i)}$ to indicate the $i$-th row of a matrix $M$ (or the $i$-th dimension of a vector) is the cleanest way and a widely accepted way, it prevented us from redefining every matrix in terms of its row vectors. If there is a more intuitive way, please let us know and we will make that change in the camera-ready version!
>
> 2. It is not clear that the manner in which the GRN is incorporated is necessary, and it is reasonable to expect that a simpler approach could yield equivalent results.
>
> This is a very reasonable concern and we’d like to clarify that, although it is not directly intuitive, the key-independent attention mechanism we introduced in section 2.2 and tested in section 3.3 is actually the generic approach to incorporate GRN – it is essentially equivalent to GLUE (Cao & Gao, 2022) where one simply encode the original graph representation with a few layers of GNN (with the last activation layer being softmax) and then take its dot product with the MLP latent. It is one of the simplest approaches to robustly incorporate GRN into the VCI framework. We note that it indeed yields similar results to the key-dependent attention when using refined GRN but key-dependent attention is more prone to GRN quality as discussed in section 3.1.
>
> Besides, we have also tried another generic approach that incorporates GRN using GNN only – integrating gene expressions into graph nodes the way nodes are constructed in section 3.2, and do node-level prediction. We tried using counterfactual-related losses and features and not using counterfactual-related losses and features, but this approach underperformed in both cases.
>
> 3. In Figure 5, the predicted distributions do not often look like the ground-truth distributions. Moreover, many of the ground-truth distributions look like control distributions, and so would be difficult to distinguish from the background anyway.
>
> This is also a really important point that we want to clarify. First we want to reiterate that, different from traditional causal inference, one is not able to give individualized supervision (other than self-supervision) for every subject since the outcomes have to be used as inputs. As can be seen from our objective construction, the counterfactual outcome is supervised by (an estimation of) distribution $p(Y’ | X, T’)$ – this intuitively means that we require each point in the predicted distribution (green) to have a reasonably high likelihood of coming from the true distributions (blue). Now given this logic, it won’t be hard to figure out why there is no guarantee that the green distribution as a whole would look like the blue distribution, yet you can generally expect the shift direction to be correct. Note that predictions are made using the control samples as factual inputs, i.e. there is a one-to-one mapping between points in green distribution and points in orange distribution. As discussed, the diversity/variation/individuality comes from the features embedded in the factual outcomes (oranges), hence the variation of the orange distribution is a determining factor of the variation of the counterfactual predictions, i.e. the green distribution. We note that when the true shift isn’t actually available, i.e. there isn’t a one-to-one mapping from the orange distribution to the blue distribution (they are two groups of different subjects), this is always going to be a challenge for the non-traditional causal inference setting. We could conduct further study in the future to see if it’d be a good idea to implement batch-wise supervision – to implement a loss that encourage the empirical distribution of a batch of counterfactual constructions (i.e. a collective group of points in the green distribution) to resemble $p(Y’ | X, T’)$ (i.e. the blue distribution).

---

### Decision · Program_Chairs · 2023-01-20

**Decision:**

Accept: poster

**Justification For Why Not Higher Score:**

The paper can be made easier to follow and its contributions more clearly distinguished from prior art.

**Justification For Why Not Lower Score:**

It could be a lower score with an encouragement to address the points raised above but overall it is a sufficient contribution for ICLR.

**Metareview: Summary, Strengths And Weaknesses:**

The paper is about causal variational Bayesian inference for single cell gene regulation data using prior knowledge about gene regulation networks.

This is a much studied area of research with a lot of prior relevant work as also pointed out by the referees.

Despite that, all reviewers are of the opinion that the paper contains novel contributions. Therefore a acceptance is the overall consensus.

**Note From Pc:**

if the above contains the word "oral" or "spotlight" please see: "oral" presentation means -> notable-top-5% and "spotlight" means -> notable-top-25%. As stated in our emails, we are disassociating presentation type from AC recommendations